# A Methodology to Estimate Functional Vulnerability Using Floating Car Data

Federico Karagulian * , Gaetano Valenti , Carlo Liberto and Matteo Corazza

ENEA Research Center Casaccia, Via Anguillarese 301, 00123 Rome, Italy
*   Correspondence: federico.karagulian@enea.it

**Abstract:** In this work, a new methodology to estimate the functional vulnerability of the road network of the city of Catania (Italy) is developed with the purpose to improve the resilience of urban transport during critical events. While the traditional approach for the estimation of vulnerability is based on topological data, the proposed methodology is based on spatial-temporal mobility profiles obtained with floating car data (FCD). The algorithm developed for the estimation of vulnerability combines topological properties of the road network with mobility patterns obtained from FCD to evaluate the consequences of failure events on trajectories and their associated travel times. The core operation of the algorithm is based on the computation of all possible travel paths within their assigned geographical zone every time a road link is disrupted. The procedure may prove useful to evaluate wide failure events and to facilitate emergency plans.

**Keywords:** vulnerability; traffic; floating car data

## 1. Introduction

In the last decade, our cities have often faced emergency situations linked to extreme anthropological or natural events. In these circumstances, the failure and disruption of critical infrastructures may have a negative impact on business activities with consequent economical losses or cause hazardous situations for territory inhabitants.

Currently, road transportation plays a fundamental role both in the development of the economy and to maintain a good quality of life in cities. Quantitative evaluations of reliability, vulnerability, and resilience are attracting attention in the planning of road networks and in transportation services for traffic congestion management.

For this purpose, the development of effective computational procedures to target critical and important infrastructures in a road network is of great interest to prevent and manage hazardous events, and therefore to minimize negative impacts on economy.

The calculation of functional vulnerability indicators allows the definition of interventions to mitigate uncomfortable situations during traffic disruptions. These indicators are associated to each edge of the road network or to city zones and represent the increased cost in mobility due to the disruption of one or more edges.

In the past, the study of vulnerability and resilience of a road network was mainly based on reliability criteria of travelling times, connectivity of the road network, and safe road traffic. Recent works focus to the analysis of the consequences of extreme weather events, climate change, and infrastructural failure on the performance of transport systems [1–3].

Vulnerability analysis of a road network arises from the need to consider the socio-economic performance and impact of degraded networks. Disruptions of most congested networks are often linked to accidental events even of short time duration, but always with a strong social and economic impact because of large traffic flows arising from the lack of proper alternative paths. In these situations, the construction of new links could strengthen the road network.

A recent OECD report highlighted the importance of the vulnerability of road networks and its effects on the performance of transport systems as an increasing consequence of the more frequent meteorological extreme events associated to climate change [4].

In the literature, vulnerability is defined according to two distinct approaches. The first one is based on the definition of analytical procedures to identify the reliability of the links and the weak elements in the network. The second one introduces measures of the disutility resulting from the disruption of a road, such as the increase of traveling time, traveling distance, and the increased economic cost of the transport.

Based on these two approaches, four main methods for the analysis of the vulnerability can be identified:

(a). The topological method considers the interconnections of a road network to identify critical paths or nodes whose disruptions would have a significant impact on the viability. The betweenness centrality and the closeness centrality are among the most important examples of topological methods [5–8]. Briefly, the betweenness centrality estimates the influence of a node with respect to the adjacent ones, while the closeness centrality is a measure of the shortest time needed to reach adjacent nodes by means of the estimation of shortest paths.

(b). The risk-based method considers the whole road network and the whole transport system, identifying the roads with the higher risk of failure due to traffic and natural events.

(c). The accessibility-based methods consider the travel demand referred to the capability of the population of a region to keep carrying on their economic and social activities when the transport network presents a failure. In the transport planning, the concept of accessibility can be defined as the level of ease access to socio-economic activity from different geographical locations [9,10].

(d). The serviceability-based methods consider the capability of a transport network to satisfy its functionality in different conditions, including the decay or failure of some edges of the network. A simple example of vulnerability analysis using the serviceability approach is the estimation of the variation of travelling times within a network upon failure of specific edges [9].

In this work, we present a serviceability-based algorithm aimed at estimating vulnerability using a dataset of floating car data available over the road network of the Italian city of Catania, Sicily. Results are presented through statistical outputs and maps which highlight vulnerable roads within the considered geographical zones. The methodology aims at being a tool for policy makers for the allocation of financial resources addressed to maintenance interventions, prevention, and effective emergency management.

## 2. Data Setups

The study area chosen for the evaluation of road vulnerability covers the province of Catania (Italy). A digitalized graph was used to represent the road network, while traffic counts were gathered from fixed sensors and floating car data (FCD). Both datasets included private and commercial vehicles.

The road network data was downloaded from OpenStreetMap [11] and covers a wide area around the city of Catania. For the purpose of this study, only the "driving" network was considered. The graph consists of approximately 74,000 nodes and 172,000 edges for a total length of 29,000 km.

FCD represents GPS time series collected by moving vehicles through an on-board terminal [12]. Therefore, it has been possible to estimate traffic volumes, speeds, traveling times, origin and destination of trajectories, and the choice of the trajectory of each trip.

FCD data were recorded during the months of February, May, August, and November 2019 in order to be representative of the four seasons of the year. These data were provided by VEM-Solutions [13]. FCD is characterized by the presence of an anonymized identification, a date and time, a progressive distance travelled by each vehicle, a code 0/1 indicating

the engine status on/off, and the instant speed. Preprocessing of FCD included the removal of outliers and the reorganization of the dataset by trips.

With an average sampling frequency of about 30 s for each record, FCD consisted of about 74 million processed traces referred to 27,000 vehicles, of which 97% were private vehicles and 3% were commercial vehicles.

On average 17% of the weekly distances were travelled during the days of Thursday, Friday and Saturday with a mean travelled distance of about 8 km for each trip. Longer travelled distances were observed on a Sunday and during the month of August. Finally, the typical travelling time was within 5 and 20 min for approximately 75% of the analyzed vehicles.

The identification of trajectories travelled by FCD traces within the road network of Catania was possible through the use of a map-matching algorithm developed in a recent work [14] that associates a sequence of GPS traces to a real path on the network. The methodology is based on a Markovian process guided by the travelled distance between traces, the quality of the signal and, the direction of the trajectory. The final result is represented by a sequence of nodes and edges crossed by consecutive traces. The accuracy of the algorithm, represented by the percentage of the correctly mapped trajectories compared to its total number, is approximately 85%. Figure 1 shows the traffic volumes obtained from map-matching of FCD on the graph for the weekday of 24 February 2019. Traffic volumes were estimated considering the total number of paths crossing the edges.

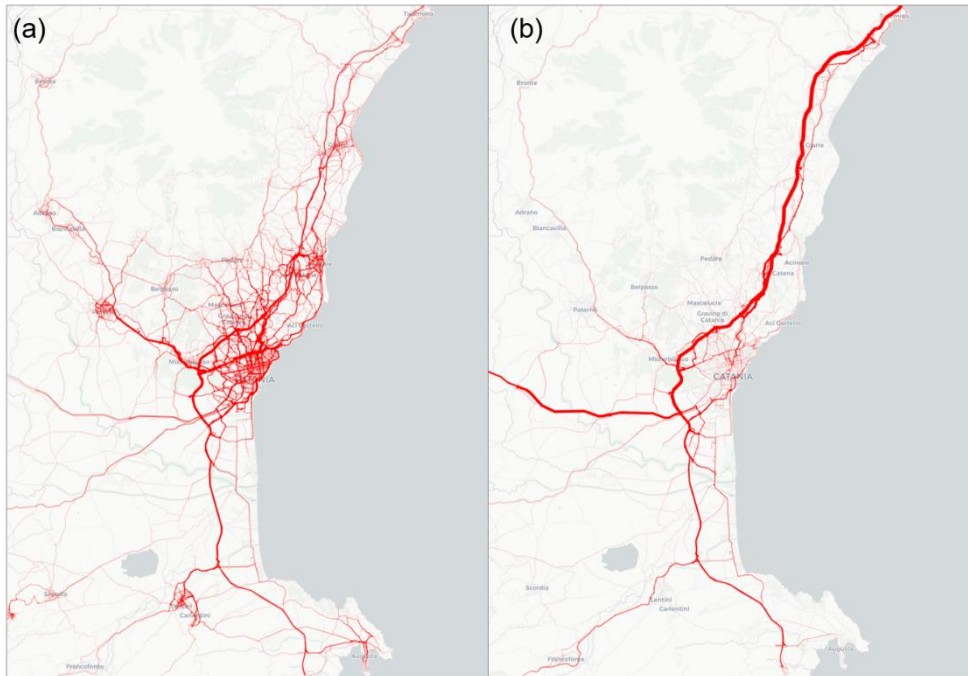

**Figure 1.** Traffic count obtained from the map matching of FCD with the road network around the area of the city of Catania, Italy for the day of 24 February 2019. (**a**) for private and (**b**) fleet vehicles. Thickness of roads is proportional to the number of times (counts) each edge was crossed during the matching process.

As shown in Figure 1a, private vehicles were mostly travelling towards the urban area of Catania, along the west Ring Road and the freeway Catania-Paternò. Instead, commercial vehicles were mainly travelling along the motorway Messina-Catania and Palermo-Catania Figure 1b).

The number of vehicles crossing each edge has been compared with the in-situ measurements carried out by the National Autonomous Roads Corporation (Azienda Nazionale Autonoma delle Strade, ANAS) at fixed monitoring stations. The comparison was carried out using an historical time series of data recorded along several road sections in the area

of Catania. Both in-situ data and map-matched FCD were aggregated by date, time, and travelling direction (ascending and descending). The hourly trend of the number of vehicles recorded in situ and obtained with FCD are shown in Figure 2. On average, the maximum number of cars was observed between 12:00 and 17:00. This value represents an hourly average calculated on several road sections during the months of February, May, August, and October 2019.

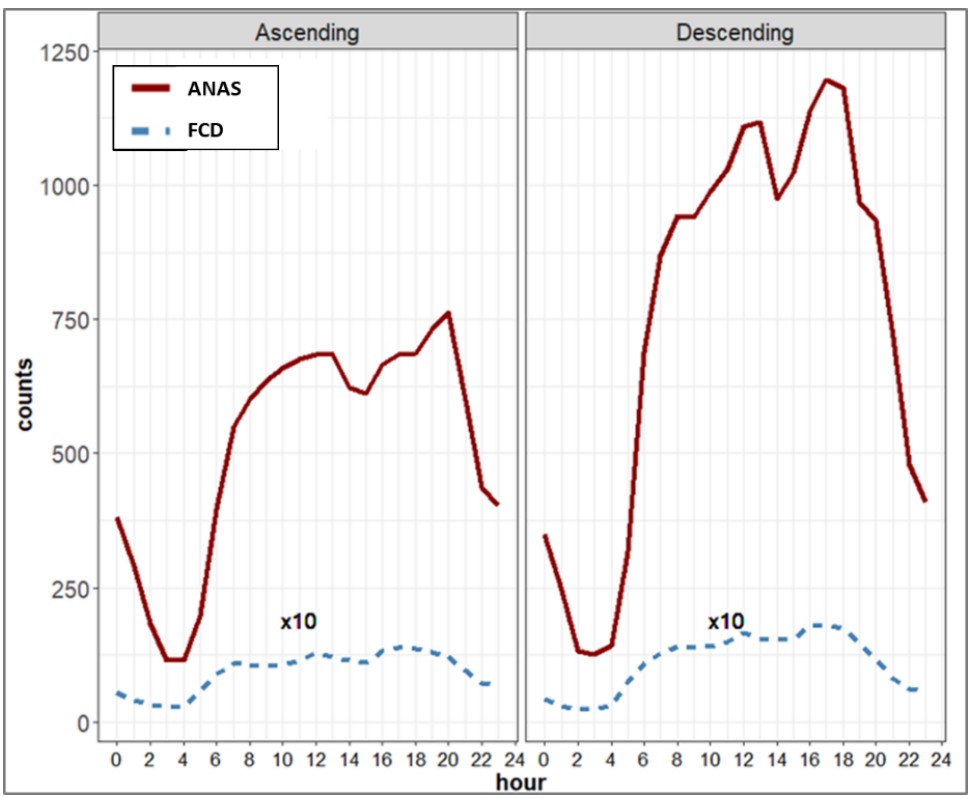

**Figure 2.** Hourly trend of the number of vehicles travelling along selected road sections in the area of Catania during the months of February, May, August and November 2019 for two travelling directions. Dashed line is referred to map-matched FCD (from VIASAT) while solid line represents in situ measurements (from ANAS, see text).

The ratio between the hourly counts of FCD and the counts obtained from in situ data along several road sections was used to estimate the penetration coefficient of the fleet vehicles equipped with GPS sensors. On average, from 08.00 to 20:00, the penetration coefficient ranges from 1.2% and 1.6% with a confidence interval of about 0.4%.

### 3. Hardware and Software

The map matching [14] and vulnerability algorithms run on a standard server (Intel®, Xenon®, Gold 6142 CPU®, 2.59 GHz processor, 64 CPUs, 128 GB RAM). Outputs are stored in a PostgreSQL data base. The map matching and vulnerability algorithms were written entirely with the open-source programming language Python using the modules networkx 2.8.8, and osmnx 1.2.3 to process road network data. Because of the high number of trips to be processed, parallel processing was setup using the Python module multiprocessing which allowed simultaneous treatment of different vehicles together with their trips. Overall, approximately 3.4 million trips in about 20 days were processed over the area of Catania.

## 4. Serviceability Analysis

The vulnerability analysis of a road network requires a priori knowledge of its topology and traffic flow (vehicles/h) obtained from map matching. For the estimation of the traffic flow, we considered the equivalent hourly rate at which vehicles crossed each edge during a 15 min interval [15]. Figure 3 shows that the highest traffic flow values are mainly observed along motorways, along the main road crossing the center of the city, and on the connecting links between motorways and primary roads or highways. The typical flow along these edges is of about 1000 vehicles/h. Further statistical analysis shows that 80% of the traffic counts has a flow of about 150 vehicles/h while only 10% can be attributed to high traffic flow, ranging from 400 to 1300 vehicles/h and mainly occurring on highways and motorways (Figure 4a). On the other hand, the speed distribution of vehicles crossing each edge showed that approximately 40% of the observed vehicles travel with a speed ranging from 20 to 40 Km/h (Figure 4b).

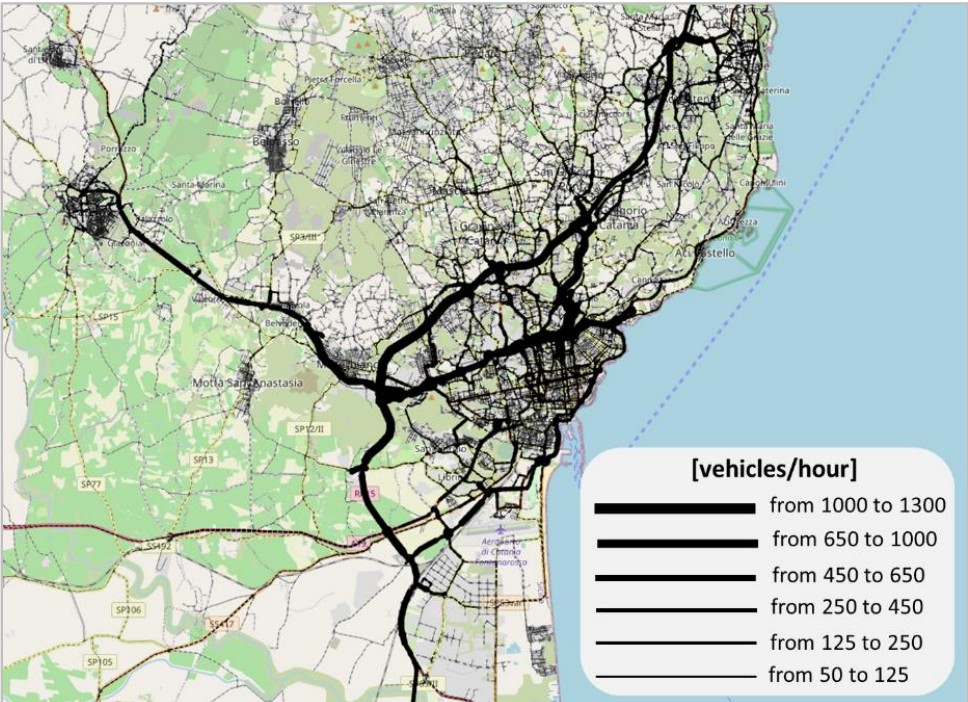

**Figure 3.** Traffic flow on the road network around the area of the city of Catania in Italy. Flow has been obtained from FCD data matched to the network and averaged over each edged during the months of February, May, August and November 2019.

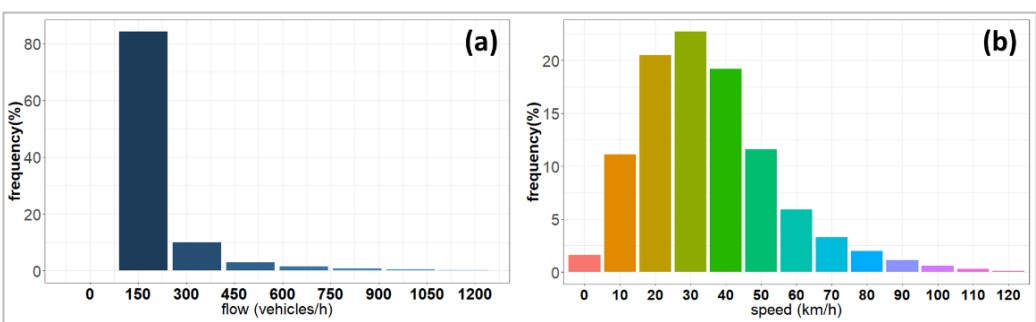

**Figure 4.** (**a**) Distribution of traffic flow obtained from FCD data referred to the month of February, May, August and November 2019. (**b**) Distribution of mean speed calculated on each edge of the road network. Elaborations are referred to the month of February, May, August and November 2019.

This result is consistent with statistics about the mean trip speed which ranges from 20 to 40 km/h for about 80% of private vehicles and 40% of the total traffic counts, whereas from 30 to 80 km/h for about 80% of commercial vehicles. As shown in Figure 1, commercial vehicles mainly travel on motorways, so their speed up to 80 km/h is expected.

Traffic flow may experience temporary variations affecting network performance in terms of capacity, traveling time, and delays. Indeed, drivers usually choose travel paths considering the shortest length or the shortest travelling time. Hence, when a disruption occurs, or when there is an increase of travel demand, the traffic outflow becomes unstable with the consequent formation of queues and loss of functionality of the network.

Congestion occurs when traffic flow is larger than the outflow capacity of the downstream road or junction. The main consequence of congestion is the formation of queues together with very low outflow speeds. To evaluate the loss of functionality of the Catania road network, a methodology to identify and quantify different types of congestion was developed. This methodology considers the number of trips and the travel time along each edge and classifies the magnitude of the congestion through the estimation of a congestion index (Figure 5) [16].

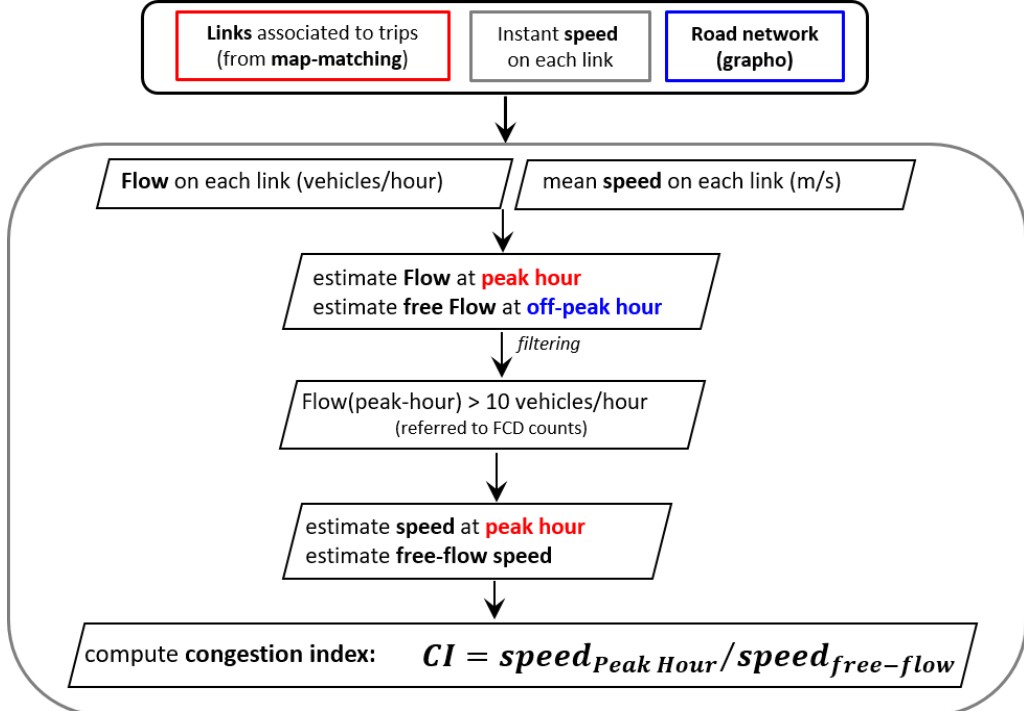

**Figure 5.** Main workflow used for the estimation of the congestion index.

The congestion index shown in Figure 5 requires the estimation of traffic at free-flow and traffic at peak hours' conditions. Free-flow has been assumed to be the minimum flow occurring on each edge during the observation time. The congestion index is defined as the ratio between speed at peak hours $\left(speed_{peak\ Hour}\right)$ and free-flow speed $\left(speed_{free-flow}\right)$ according to the following equation:

$$CI = \frac{speed_{Peak\ Hour}}{speed_{free-flow}} \tag{1}$$

The free-flow speed is estimated on hourly basis at the lowest flow recorded on each edge. The speed at peak hours is estimated at the highest flow recorded on each edge. As shown in Figure 6, in the condition of peak hours, the largest number of vehicles crossing

each edge was recorded in the morning from 08:00 to 09:00 and in the evening from 18:00 to 19:00.

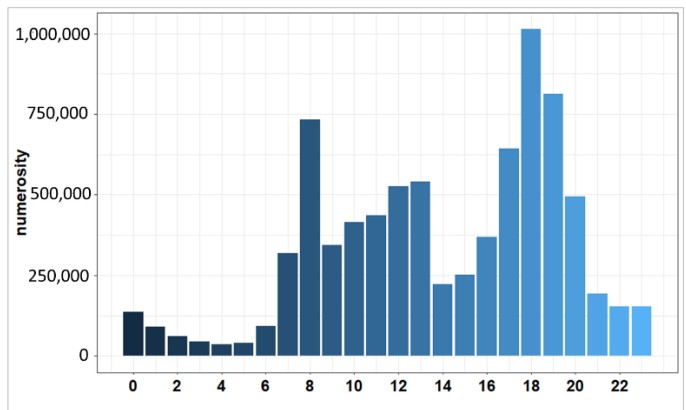

**Figure 6.** Hourly distribution of number of vehicles crossing each edge at peak hours' conditions. Observations are referred to the months of February, May, August and November 2019.

The congestion index shows a remarkable seasonal variation. Indeed, during the winter month of February, when commercial activities occur, the most congested edges are found in urban areas (Figure 7). On the other hand, during the summer month of August, when commercial activities are reduced, congested edges are mainly found along roads connecting touristic places of relevant interest. However, the congestion index referring to highways and motorway does not show substantial differences between the two seasons.

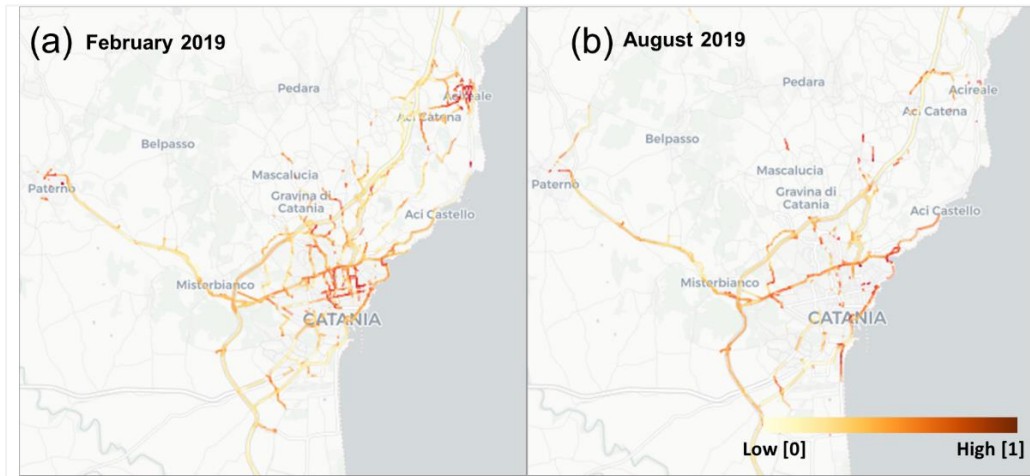

**Figure 7.** Snapshots of congestion index map estimated in the road network around the city of Catania (Italy) for the months of February and August 2019.

## 5. Vulnerability Analysis

For the evaluation of vulnerability, we have used two methodologies, the first one based on topology, the second one on the evaluation of risk through the identification of the edges with higher risk of failure.

### 5.1. Topological Vulnerability

Two well-known topological methods to measure the importance of each node and its influence on the movements within the road network are the betweenness centrality and the closeness centrality [7,8,17]. A first elaboration of vulnerability was carried out using the betweenness centrality model to estimate the centrality of each node within a road network. The betweenness centrality is based on the calculation of the shortest paths

in a connected Graph [5,6] and determines the influence of a node compared to the traffic flow between each pair of nodes or adjacent edges. Usually, the betweenness centrality of a node $x$ is defined as:

$$BC(x) = \sum_{u \neq v \neq x} \frac{shortest\_path(u,v|x)}{shortest\_path(u,v)} \tag{2}$$

where the $shortest\_path(u,v)$ considers the total number of shortest paths from node $u$ to node $v$, while the $shortest\_path(u,v|x)$ considers the shortest paths from $u$ to $v$ crossing node $x$. The calculation of the shortest path has been carried out using the Dijkstra's algorithm that, from a given graph and a source vertex in it, finds all possible shortest paths from source to all vertices [18]. The sum performed in Equation (2) refers to the whole number of edges of the graph. Shortest paths are determined by minimizing the total length of all road links within the path from node $u$ to node $v$. For the analysis of the Catania's network, each edge has been weighted with its average travel time or cost obtained from the elaboration of FCD data. This has been accomplished by estimating the mean travel speed on each edge using the map-matching algorithm. The travel time has been then calculated using the length of the edge. The FCD data provide the advantage of a real measure of the vehicle speed that can substitute the maximum speed provided by Open Street map. In addition, it is reasonable to suppose that not all the vehicles circulating on the street are travelling at the maximum speed limit. This highlights the importance of using realistic measures (FCD data). As presented in Figure 8, the normalized (0–1 range) value of betweenness centrality shows differences between the weighted and unweighted network. One of these is the assignation of a not-negligible value to the freeway Catania-Paternò (Figure 8) that does not assume any importance when considering the unweighted network. However, the betweenness centrality only highlights the fast-flowing road network (i.e., motorways and highways) without giving information about the importance of urban roads.

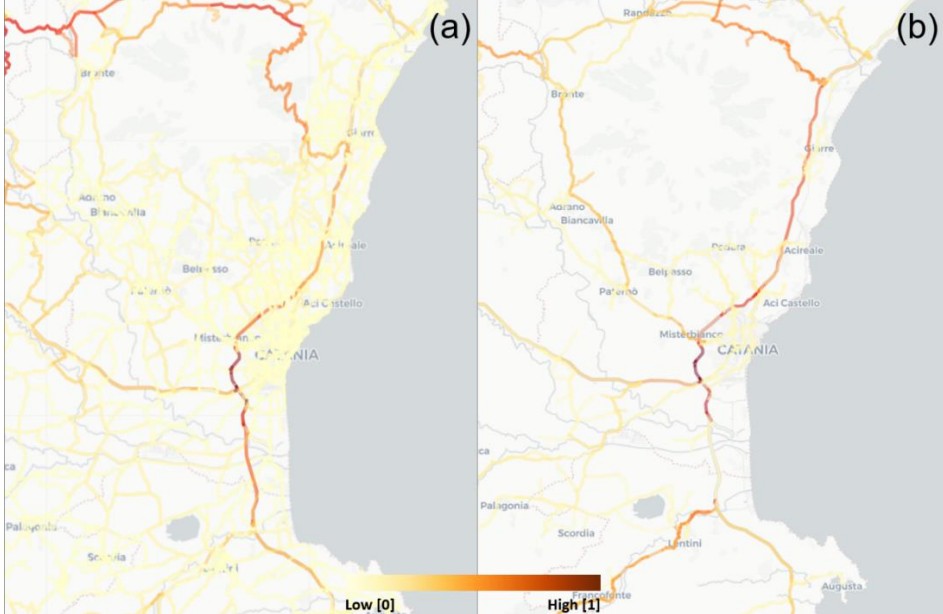

**Figure 8.** Normalized Betweenness centrality of the road network in the province of Catania during the month of August 2019. (**a**) unweighted network, (**b**) weighted network using travel times.

Another parameter largely used to characterize topological vulnerability is the closeness centrality. This method is based on the estimation of the length of shortest paths between a given node and all other nodes in the network [19]. The closeness centrality is

defined as the reciprocal of the sum of the shortest path lengths between a node and all other nodes:

$$CC(x) = \frac{N-1}{\sum_{u \neq x} shortest\_path(u,x)} \tag{3}$$

where the $\sum shortest\_path(u,x)$ is the sum of all possible shortest paths between the node $u$ and node $x$ across the entire network. The closeness centrality characterizes a connected network by quantifying the spatial accessibility of a node. It can also be considered an indicator of the reachability of nodes, through the estimation of the shortest paths. As shown in Figure 9, highest values of closeness centrality are observed along the West Ring Road of the motorway Palermo-Catania and the motorway Catania-Siracusa. Indeed, these roads represent the faster path to reach the center of the city of Catania when coming from peripheral zones. On the other hand, Figure 9 shows that a large fraction of the roads in the center of the city is characterized by the lowest closeness centrality. This indicates the difficulty to reach several neighborhoods in the city center with a consequent reduction of the traffic flow in these zones. It is once again important to stress that this behavior is based on real traffic speeds observed on the considered edges, retrieved from FCD data.

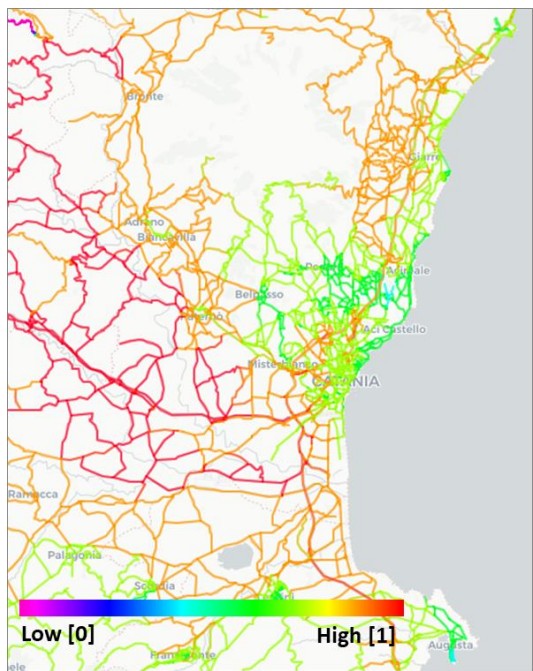

**Figure 9.** Closeness centrality of the road network around the city of Catania during the month of August 2019.

### 5.2. Vulnerability Based on Risk and Functionality

The other approach used for the calculation of vulnerability is the risk-based method. This approach consists in the quantification of the impact a road closure has on the whole network, rather than the evaluation of the reduction of the functionality of the edge itself by simulating slowdown of traffic flow. For this purpose, each link has been considered as an interconnected edge within a geographical zone represented by a hexagonal cell. For this reason, tessellation or zoning has been carried out in order to better represent the interconnection between edges and to highlight the effect of potential road closure within a given zone. Assuming a probability of 1 for a disrupted link and a probability of 0 for a connected link, the vulnerability of each geographical cell can be defined as the sum of importance of each link within the cell:

$$vulnerability_{[cell]} \sim \sum_{i}^{N_{links}} importance_i \tag{4}$$

The *importance* represents the time (defined as vehicle × hour) lost by all drivers when a link is disrupted. Basically, the importance of a link is strictly related to the number of vehicles crossing that link and therefore to the flux of vehicles. More generally, if the number of drivers travelling from an origin towards a destination (OD) is known, it is possible to estimate the frequency at which an edge is crossed and therefore the amount of time lost along the route.

For the estimation of vulnerability along the Catania network we have followed the methodology proposed by the work of Jenelius and Mattson [20], assuming that the consequences derived by the disruption of a set of links within a cell are an indication of their importance within the network. The vulnerability algorithm has been setup according to the flow diagram illustrated in Figure 10. Outputs from map matching have been used to estimate the traffic counts, the mean speed occurring on each link and, to find all possible Origin-Destination trajectories along the network for a given observation time.

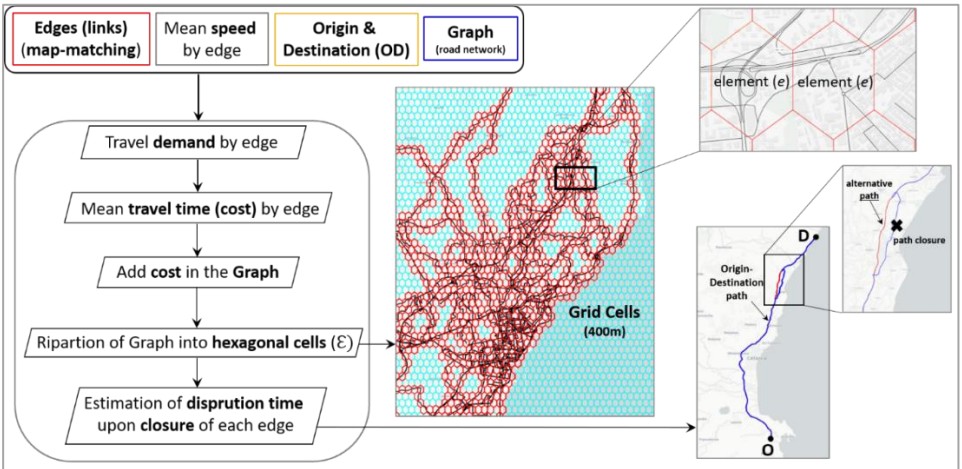

**Figure 10.** Main workflow used in the process to assess vulnerability of a road network.

As reported above, we operated a tessellation or zoning of the road network using hexagonal cells of 400 m diameter (the diameter is referred to the circumscribed circumference, corresponding to the double of the hexagon side). Overall, the whole network has been zoned with approximately 30,000 cells (Figure 10).

As shown in Figure 10, each cell includes a limited number of connected links. If one of those links get disrupted, other links in the same cell will also get congested or will suffer long delays. Ideally each cell should contain only few interconnected links whose ending nodes are located in the adjacent cells. In this way, it is possible to model the vulnerability of each link by considering the effect of the adjacent cells according to the following procedure:

(1)　For each edge in the cell, origins and destinations are assigned to the associated trip involving the edge.

(2)　A travel cost (time) is assigned to each edge in the network. The travel cost is calculated as the ratio between the mean speed along each edge and its length. The mean speed of each edge is computed from the average of all instant speeds recorded from all FCD involving the edge.

(3)　In the connected scenario, the shortest and faster path between each origin and destination node is calculated assuming none of the links within the cell is disrupted.

(4)　In the disrupted scenario, a penalty time (defined as the duration of the disruption) is added to the travel cost of each edge and the shortest path between each origin and destination is computed again. For this study, the penalty time was set at a value of 5 h.

(5)　For each route crossing the cell, the difference between the travel time in a connected scenario and in a disrupted scenario is calculated considering all possible combi-

nations between each origin and destination. These differences were then added together to estimate the time lost on each link as well as in each cell.

Therefore, as stated above, the *importance* of a link is represented by the time lost during its disruption and can be defined according to the following Equation (5) [20]:

$$importance_{i,j}^e = \begin{cases} Flux_{i,j} \times \Delta t_{i,j}^e \left( penalty - \frac{\Delta t_{i,j}^e}{2} \right) & if\ \Delta t_{i,j}^e < penalty \\ \frac{Flux_{i,j} \times penalty^2}{2} & otherwise \end{cases} \tag{5}$$

where $Flux_{[OD]}$ is the travel demand that can be approximated with the flux of vehicles travelling from one origin to one destination and, $\Delta t_{link}^{OD}$ is the delay time accumulated by the driver on each link within the cell during the disruption time window.

For the estimation of vulnerability over zones represented by cell, it is important to stress that each link may cross multiple cells. In addition, considering that within each cell links are connected, the disruption of a link might affect the congestion of all links within that cell [21]. For this reason, cell vulnerability has been estimated by considering the highest vulnerability value among the links crossing that cell. Figure 11 shows the pseudo code used for the computation of the delay time used for the estimation of the vulnerability. Basically, the algorithm computes all possible travel paths obtained every time a link is disrupted within its assigned cell.

---

**Input**: *Graph*, set of *links* in each cell $\mathcal{E} = \bigcup_i^{NCELLS} e_i$, *Origin & Destination* $(i,j) \in N_{OD}$,
     penalty time $\tau$
**Output**: total *DELAY* time $\Delta T_{OD}^e$ for each *link* $\in e$

---

```
for each e in Ɛ do
    DELAY = 0
    Let DELAY_CELL[ ] store the total delay time within every cell
    select OD ∈ e
    for each (i,j) in OD
```
*Find time of the shortest path based on the "cost" (time) in the NULL scenario*
```
        Compute SP⁰₍ᵢ,ⱼ₎ = ShortestPath(i, j, weigth = "cost")
        Compute t⁰₍ᵢ,ⱼ₎ = TimeShortestPath(i, j, weigth = "cost")
            for each link in SP⁰₍ᵢ,ⱼ₎
                Compute travel demand x⁰₍ᵢ,ⱼ₎ = Σₖᴺˡⁱⁿᵏˢ flowₖ
```
*Close each link in the "element" by assigning a penalty time (disruption) and recalculate shortest path*
```
        for each link k in e
            travel time = "cost" + τ
        Compute SP^e₍ᵢ,ⱼ₎ = ShortestPath(i, j, weigth = "travel time")
        Compute t^e₍ᵢ,ⱼ₎ = TimeShortestPath(i, j, weigth = "travel time")
```
*Calculate "closure impact" for each OD*
```
        Δt^e₍ᵢ,ⱼ₎ = t^e₍ᵢ,ⱼ₎ − t⁰₍ᵢ,ⱼ₎
        if Δt^e₍ᵢ,ⱼ₎ < τ
            DELAY = DELAY + Δt^e₍ᵢ,ⱼ₎ · x⁰₍ᵢ,ⱼ₎ (τ − Δt^e₍ᵢ,ⱼ₎/2)
        else
            DELAY = DELAY + (x⁰₍ᵢ,ⱼ₎ × τ²)/2
```
*Restore initial travel time*
```
        for each link k in e
            travel time = "cost"
```

---

**Figure 11.** Pseudo code of the Vulnerability algorithm developed for the case study (see text for more details).

Elaborations

Vulnerability analysis along the Catania network was estimated for the months of February and August 2019 as representative of the winter and summer season, respectively. The vulnerability of each link and cell was calculated by considering all trips and traffic flows computed for both the months of February and August 2019. Map-matching carried out on each edge resulted in the estimation of fundamental parameters, such as mean speed, travel time, traffic count, and traffic flow. As final output, vulnerability values were normalized to the highest value found in the network. Therefore, the vulnerability of each link was expressed with a value between 0 (minimum) and 1 (maximum).

It can be observed that only few links and cells are characterized by high vulnerability values (Figure 12). Indeed, most of the links in residential areas show very low values, whereas high values are observed along motorways, connecting links and along the main primary road crossing the center of the city. The vulnerability of the Catania network does not show substantial differences between the winter (February) and summer (August) seasons, with the exception of few roads along the north side of the main motorway, in the urban area of Catania, and of the town of Acireale that show higher vulnerability values during the summer compared to the winter. However, the fact that vulnerability does not show considerable differences between the winter and the summer along most of the roads can be an indication of the intrinsic nature of vulnerability. Indeed, although we have used FCD data corresponding to two different periods of the year, most of the roads show the same vulnerability value. Further analysis shown in Figure 13 highlights links with vulnerability values larger than 0.5. In that case, it is observed that vulnerability mainly increases along links with high traffic flows and along the intersections of primary roads where highways and ramps are present. Most critical links can be targeted and can undergo further analysis. For instance, links with high vulnerability values can be better investigated through an interdependency analysis to understand the breakdown of traffic flow among adjacent links. Another analysis could involve the evaluation of travelling time of all trips crossing critical links. In other words, we stress the importance of targeting vulnerable links as the starting points of further analysis to investigate possible solutions to mitigate the vulnerability of the road network [22,23].

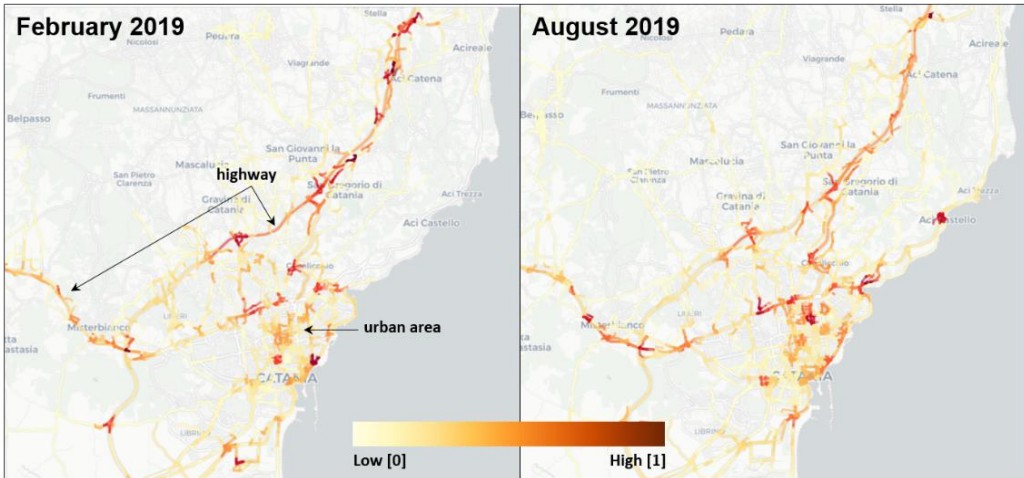

**Figure 12.** Snapshots of vulnerability estimated in the road network around the city of Catania (Italy) for the months of February and August 2019. Links within a 400 m cell are colored according to their vulnerability value. When a link crosses multiples cells, the vulnerability is given by the sum of the cumulative values obtained for each Origin-Destination.

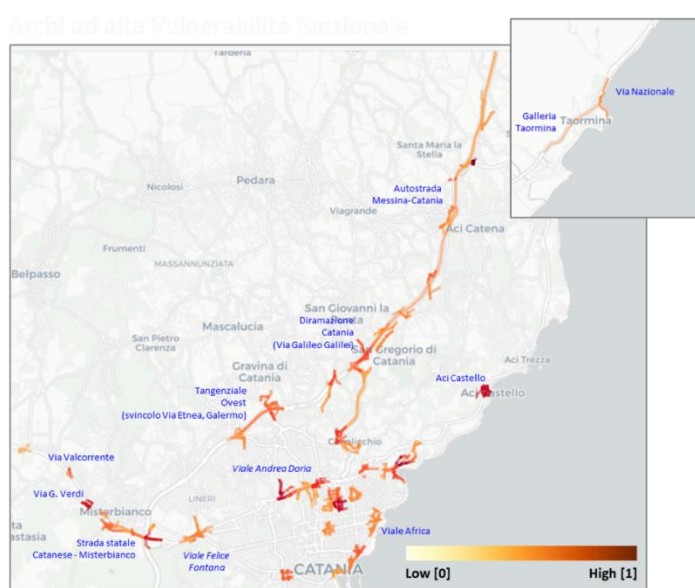

**Figure 13.** Road link in the Catania network with vulnerability larger than 0.5 (August 2019).

The interconnectivity of links within a given zone or cell results in the assignation of the highest vulnerability value that can be observed in that cell. Vulnerability analysis by zones or cells shows that vulnerability becomes significant when a particular event affects a given area (Figure 14). This event may be represented by conditions of intense traffic flow or by a natural event such as a flooding or a collapse of the road. Figure 14 shows that all the zones crossed by the main motorway towards the city of Catania have not-negligible vulnerability values. The same situation was observed in most of the zones around the center of the city. In this case, as shown in Figure 13, although only a few links are associated to high vulnerability, the interconnection between links clearly affects other neighbor roads and therefore the zones they cross [24]. For this reason, zoning of the vulnerability may be useful to setup possible emergency plans based on the value of importance estimated for each zone (cell) as well as to assign a priority in the interventions over a given zone.

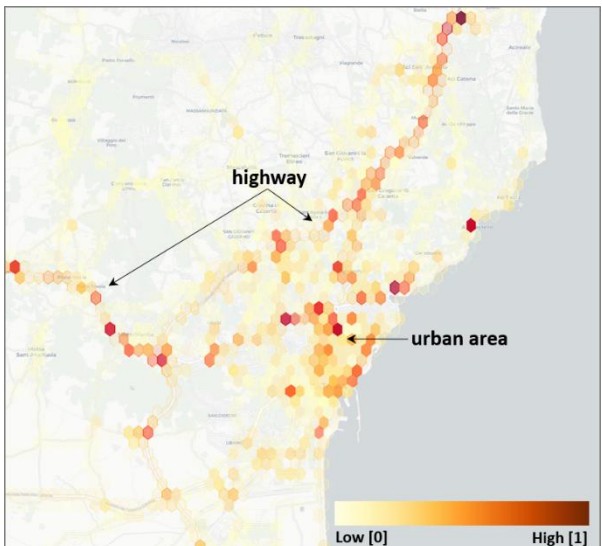

**Figure 14.** Map of the vulnerability by zone (or cell). To each cell the maximum vulnerability value of the edge crossing it has been assigned. Cells are colored according to their maximum vulnerability value.

## 6. Conclusions

In this work, a methodology to estimate road vulnerability using map matched FCD data has been proposed, together with a pseudo code illustrating the main steps for its practical implementation. This methodology showed that it is possible to identify roads and geographical zones that may require interventions to minimize the impact related to disruptions that, in most cases, is represented by traffic congestion or disasters caused by natural phenomena [21].

It is shown that, while the traditional vulnerability methods based on topology assign importance to road links only considering their rank within the network, the vulnerability risk-based method provides information along the whole network by considering the change in travel demand. This was possible because FCD data allowed the estimation of travel speed, travel time, and traffic flow over the whole road network.

The case study illustrated in this work shows that roads and zones with high vulnerability are both located along motorways/highways and in the center of the city where most people live and economic activity occurs on daily basis (Figure 13). This is extremely important when considering emergency plans to be implemented before road disruption occurs. Finally, the outcomes of this work may improve emergency plans in elaborating the strategies for the choice of alternative roads to be used in case of emergencies or to set priorities for the improvement of the existing infrastructure in order to avoid congestion.

Future studies can be focused on a more detailed analysis of the most critical links identified by the vulnerability method presented in this work. For instance, as stated in Section 5, links with high vulnerability values can be further investigated through an interdependency analysis in order to characterize the impact of the link on the traffic flow of adjacent links. Moreover, an a priori analysis of the travelling times of all trips crossing the critical considered links can be performed for different scenarios in order to elaborate detailed emergency plans.

**Author Contributions:** Conceptualization, F.K., G.V. and C.L.; data curation, F.K. and M.C.; formal analysis, F.K. and C.L.; investigation, F.K. and G.V.; methodology, F.K. and M.C.; supervision, G.V.; writing—original draft, F.K.; writing—review and editing, F.K. All authors have read and agreed to the published version of the manuscript.

**Funding:** This research received no external funding.

**Data Availability Statement:** Not applicable.

**Acknowledgments:** The research reported in this project was funded by the Ministero dell′ Università e della Ricerca through the project RAFAEL (System for Risk Analysis and Forecasting for Critical Infrastructures in the AppenninEs dorsaL Regions).

**Conflicts of Interest:** The authors declare no conflict of interest.

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
