# Peer review of "A Methodology to Estimate Functional Vulnerability Using Floating Car Data"

_sustainability, doi:10.3390/su15010711_

Round 1

Reviewer 1 Report

As per the revised document, 

1. How the free flow speed data was collected is not clear?

2. There was no traffic information presented for the entire network or major part of it. It will be better to provide such kind of speed and flow data. Fig. 2 information is not sufficient.

3. Any latest data is available for analysis? Figure 8 shows 2019 data.

4.  How many zones in the city road network and how many links and nodes are not clear from the manuscript?

5. Conclusions are too general in nature.

Author Response

Reviewer 1

1.How the free flow speed data was collected is not clear?

Free-flow was assumed to be the minimum flow occurring on each edge during the observation time. Amendments were done into the text (in particular in Section 2 and Section 4) to explain how the free-flow and the free-flow speed have been collected.

2. There was no traffic information presented for the entire network or major part of it. It will be better to provide such kind of speed and flow data. Fig. 2 information is not sufficient.

Amendments were done into the text to illustrate traffic information together with speed and flow data. Some of this information was already reported in the Data Setups session and therefore, it has been improved with the addition of other Figures and statistical outputs.

3. Any latest data is available for analysis? Figure 8 shows 2019 data.

We do not have any data covering a later period after 2019. The input data for this work were purchased before 2019 and the work has been carried out afterwards.

4. How many zones in the city road network and how many links and nodes are not clear from the manuscript?

The whole network was zoned with about 30k cells (Figure 10). Amendments were done into the text to describe this point

5. Conclusions are too general in nature.

Conclusions have been completely rewritten with particular emphasis on the purpose of the vulnerability methodology and its added value compared to other topological methods.

Reviewer 2 Report

A new methodology to estimate the functional vulnerability of a road network has been developed with the purpose to improve the resilience of urban transport during critical events. The study has shown some positive results. The reviewer has the following comments:

1)     The abstract seems to be prolix. Please refine it thoroughly for a clear and concise statement.

2)     It is hard to follow the storyline of the introduction. More remarks should be added to display the difficulties, novelty and efficiency of the proposed results.

3)     The authors are suggested to summarize the contribution of their works in the Introduction section.

4)     For the results presented in the Figure 6, more explanations on them seem necessary and helpful to readers.

Author Response

Reviewer 2

A new methodology to estimate the functional vulnerability of a road network has been developed with the purpose to improve the resilience of urban transport during critical events. The study has shown some positive results. The reviewer has the following comments:

1. The abstract seems to be prolix. Please refine it thoroughly for a clear and concise statement.

The abstract has been simplified and made more concise to target the main purpose of the proposed methodology.

2. It is hard to follow the storyline of the introduction. More remarks should be added to display the difficulties, novelty and efficiency of the proposed results.

The introduction has been completely reorganized and rewritten in order to highlight the novelty and purpose of this work. Several redundant parts were removed and the text is now more fluid.

3. The authors are suggested to summarize the contribution of their works in the Introduction section.

The introduction has been completely reorganized and rewritten with explanation of the main results and purpose of the paper.

4. For the results presented in the Figure 6, more explanations on them seem necessary and helpful to readers.

Figure 6 (now Figure 9) has been better explained in Section 5 in order to help the reader to understand the difference between the vulnerability methods presented in the paper.

Reviewer 3 Report

In this manuscript, the authors proposed a new approach to estimate the functional vulnerability of a road network, which aims to improve the resilience of urban transport during critical events. In the proposed approach, they utilized the spatial-temporal mobility profiles obtained with Floating Car Data (FCD) to estimate the congestiion index and vulnerability of road networs, and futher combined them to those obtained with the tradtional approaches such as those using topological data and on traffic demand.

In my opinion, the idea could be intersting and it is worth the potential recommendation. However, I have also some conerns as follows

1)     The authors only consider the data in Feb, May, Aug and Nov., but I want to see the results at the end of December, which could be the most congesting hours in one year.

2)     The manuscript is not well-written or organized. I wish the authors can reorganize the whole paper to largely enhance the readability.

3)     The words and expressions need to be further polished. As an example, in Line 18, “at the data”àto date; in Line 237, “ration”àratio; in Line 270. “in a Connect Graph”àin a connected graph.

4)     References are incomplete. Regarding the node centrality, two related works [Applied Mathematics and Computation, 2018, 334: 388-400; IEEE Transactions on Systems, Man, and Cybernetics: Systems, 2021, 51(12): 7823-7837.] can be mentioned here.

Author Response

Reviewer 3

In this manuscript, the authors proposed a new approach to estimate the functional vulnerability of a road network, which aims to improve the resilience of urban transport during critical events. In the proposed approach, they utilized the spatial-temporal mobility profiles obtained with Floating Car Data (FCD) to estimate the congestion index and vulnerability of road networks, and further combined them to those obtained with the traditional approaches such as those using topological data and on traffic demand.

In my opinion, the idea could be interesting and it is worth the potential recommendation. However, I have also some concerns as follows

1. The authors only consider the data in Feb, May, Aug and Nov., but I want to see the results at the end of December, which could be the most congesting hours in one year.

Input data for this work were purchased for months that could represent the seasonal trend of FCD data during one year. Therefore, the choice of the month of February, May, August and November were intended to represent the seasons of Winter, Spring, Summer and Fall. We do not have data purchased for the month of December 2019.

2. The manuscript is not well-written or organized. I wish the authors can reorganize the whole paper to largely enhance the readability.

The manuscript has been completely reorganized and most of its part reworded in order to improve its readability. The English form has been properly checked.

3. The words and expressions need to be further polished. As an example, in Line 18, “at the data”à to date; in Line 237, “ration”àratio; in Line 270. “in a Connect Graph”àin a connected graph.

Amendments were done in the text.

4. References are incomplete. Regarding the node centrality, two related works [Applied Mathematics and Computation, 2018, 334: 388-400;  and  IEEE Transactions on Systems, Man, and Cybernetics: Systems, 2021, 51(12): 7823-7837.] can be mentioned here.

The references suggested above were added to the revised version. We thank the reviewer for this suggestion.

Round 2

Reviewer 1 Report

There was no major change in the structure of the manuscript as claimed by the authors.

More result discussion of the results is required.

Author Response

Dear Reviewer,

Thanks again for your observations and suggestions.

Below the answers to your comments:

1) There was no major change in the structure of the manuscript as claimed by the authors.

As result of the initial reviewer’s suggestions, most of the content of the paper has been modified. The work consisted in several cuts in the text as well as in new rephrased concepts that help to make the work clearer and understandable. When we claimed about a change in the structure of the manuscript, we intended in a more fluent, clear and concise explanation of the sequence of steps described in the paper in order to build an algorithm to estimate road vulnerability at road and zone level.

2) More result discussion of the results is required.

Discussion of the results has been improved with the addition of more insights about the methodology and the main results obtained from the vulnerability model. Amendments have been done in several paragraphs of Section 5 (Vulnerability analysis). Moreover, a sentence on further possible work has been added in the Conclusions.

Finally, four more references have been added to the text.

Reviewer 3 Report

The authros have made some necessary revisions, and I am willing to recommend it to be accepted in the present form.

Author Response

Considering that the Reviewer 3 was satisfied with the previous revision, we did not make any further revision. However, we made amendments to the manuscript in order to meet the requests of Reviewer 1.